

# Rashba spin-orbit coupling and nonlocal correlations in disordered 2D systems

Yongtai Li, Gour Jana and Chinedu E. Ekuma⋆

Department of Physics, Lehigh University, PA

⋆ che218@lehigh.edu

## Abstract

We present an extension of the dynamical cluster approximation (DCA) that incorporates Rashba spin-orbit coupling (SOC) to investigate the interplay between disorder, spin-orbit interaction, and nonlocal spatial correlations in disordered two-dimensional systems. By analyzing the average density of states, momentum-resolved self-energy, and return probability, we demonstrate how Rashba SOC and nonlocal correlations jointly modify single-particle properties and spin-dependent interference. The method captures key features of the symplectic universality class, including SOC-induced delocalization signatures at finite times. We benchmark the DCA results against those obtained from the numerically exact kernel polynomial method, finding good agreement. This validates the computationally efficient, mean-field-based DCA framework as a robust tool for exploring disorder, spin-orbit coupling, and nonlocal correlation effects in low-dimensional systems, and paves the way for simulating multiorbital and strongly correlated systems that were previously inaccessible due to computational limitations.



# 1 Introduction

Understanding how disorder influences electronic transport remains a foundational challenge in condensed matter physics, particularly in low-dimensional systems where quantum interference plays a dominant role [1–4]. In such systems, electron scattering from spatially distributed impurities can suppress diffusive motion entirely, leading to Anderson localization [5]. The mechanism is rooted in coherent backscattering, where constructive interference between time-reversed paths enhances the probability of electron return [1,5,6]. In three dimensions, this interference leads to a disorder-driven metal-insulator transition at a finite critical disorder strength [5]. In contrast, for systems in two or fewer dimensions ($d \leq 2$), all electronic states become localized even at infinitesimal disorder, as described by single-parameter scaling theory [7]. As a result, 2D disordered systems without additional symmetry-breaking fields fall within the orthogonal universality class and do not exhibit a true Anderson transition [7–12].

The inclusion of spin-orbit coupling (SOC), however, alters this picture by breaking spin-rotational symmetry while preserving time-reversal symmetry [1, 13–15]. This leads to the suppression of coherent interference and drives the system into the symplectic universality class, thereby restoring the possibility of an Anderson transition in 2D [8,16–19]. Crucially, this shift from the orthogonal to the symplectic universality class is the same physics responsible for weak-antilocalization: Rashba SOC suppresses coherent backscattering by locking spin to momentum. As we will show via both the average density of states and the time-resolved return probability, our DCA-SOC framework not only captures spectral shifts, but quantitatively reveals this crossover in interference behavior. The interplay between SOC and disorder has also been linked to other phenomena, including the anomalous Hall effect (AHE) driven by skew scattering [20] and the emergence of spin-orbit torques [21, 22]. In two-dimensional systems, such as $Ga_{1-x}Mn_x As$ and other materials with Rashba SOC [23–25], broken inversion symmetry introduces new scaling behavior in conductivity and modifies carrier dynamics [26–32].

While numerous studies have employed numerically exact methods such as the exact diagonalization, transfer matrix method, and kernel polynomial method (KPM) [16–18,30,33,34] to investigate electron localization in 2D disordered systems with SOC, dynamical mean-field theory (DMFT)-based approaches remain less explored. In particular, incorporating nonlocal spatial correlations, which are essential for capturing coherent backscattering effects, is a necessary step toward developing a mean-field framework capable of addressing localization physics. Cluster extensions of DMFT, such as the dynamical cluster approximation (DCA) [35, 36], offer a route toward this goal by incorporating short-range nonlocal correlations. Unlike single-site methods such as the coherent potential approximation (CPA) [37,38], DCA restores lattice symmetries and captures momentum-dependent corrections. DCA has been benchmarked against numerically exact methods and shown to provide qualitatively and quantitatively accurate results at significantly lower computational cost [39–41].

In this work, we extend the dynamical cluster approximation [35, 36] to incorporate Rashba spin-orbit coupling [23], enabling the study of its interplay with disorder on a two-dimensional square lattice. While previous studies have incorporated Rashba spin-orbit coupling (SOC) into strongly correlated systems within single-site mean-field approaches such as the DMFT and CPA [42], as well as their cluster extensions [43, 44] to investigate its role in topological superconductivity, to the best of our knowledge, no work has yet examined the competition between random disorder and Rashba SOC using a nonlocal mean-field framework. While numerically exact approaches such as the kernel polynomial method [45] have proven effective in characterizing disordered systems with SOC, their computational cost can be prohibitive for large system sizes or multiorbital models. The DCA-based framework developed here offers a computationally efficient alternative that systematically captures non-

local spatial correlations and is readily extensible to more realistic material-specific Hamiltonians [46–48]. We compute single-particle observables, including the average density of states (ADOS) and self-energy, for both single-site and finite-size clusters. Our results demonstrate that Rashba spin-orbit coupling suppresses disorder-induced momentum dependence in the self-energy, thereby reducing the spatial variation in quasiparticle scattering rates. We can probe the onset of localization through the return probability, which serves as a dynamic diagnostic of localization behavior. At finite times, the inclusion of SOC reduces return probabilities, indicating SOC-induced delocalization. However, in the infinite-time limit, the system remains delocalized for all disorder strengths, consistent with the mean-field nature of DCA, which does not capture true Anderson localization [35]. Finally, we benchmark the ADOS computed within DCA against those obtained from KPM, and find excellent agreement across a range of disorder strengths. This confirms that the extended DCA-SOC framework provides a numerically efficient and physically robust platform for studying the effects of disorder and spin-orbit coupling in low-dimensional systems.

The rest of the manuscript is organized as follows. In Sec. 2, we introduce the disordered 2D lattice model with Rashba spin-orbit coupling and outline the dynamical cluster approximation framework extended to capture spin-resolved and nonlocal disorder effects. In Sec. 3, we present a comprehensive analysis of the single-particle properties, including the average density of states, momentum-resolved self-energy, and return probability, to investigate the interplay between SOC, disorder, and spatial correlations. We further benchmark the DCA results against those obtained from the kernel polynomial method, demonstrating both the accuracy and computational advantages of our approach. In Sec. 4, we summarize our findings and outline future directions, including the integration of a typical medium framework for capturing Anderson localization and the incorporation of realistic, material-specific Hamiltonians to enable quantitative studies of disordered spin-orbit-coupled systems.

## 2 Method

We study the Anderson model on a square lattice in the presence of Rashba spin-orbit coupling. The total Hamiltonian is given by

$$\hat{H} = -\sum_{\langle i,j \rangle,\sigma} t\left(\hat{c}^{\dagger}_{i\sigma}\hat{c}_{j\sigma} + \text{H.c.}\right) + \sum_{i,\sigma} V_i \hat{c}^{\dagger}_{i\sigma}\hat{c}_{i\sigma} + \hat{H}_{\text{SO}}, \tag{1}$$

where $\hat{c}^{\dagger}_{i\sigma}$ ($\hat{c}_{i\sigma}$) is the creation (annihilation) operator for an electron at site $i$ with spin $\sigma$, and $t$ is the hopping amplitude between nearest-neighbor sites $\langle i,j \rangle$. We set $4t = 1$ as the unit of energy throughout. The on-site disorder potential $V_i$ is randomly sampled from a binary distribution,

$$P(V_i) = \frac{1}{2}\delta(V_i + W) + \frac{1}{2}\delta(V_i - W), \tag{2}$$

where $W$ characterizes the disorder strength.

The Rashba SOC term in momentum space is given by

$$\hat{H}_{\text{SO}} = \sum_{\mathbf{k}} \alpha_{\text{SO}}(\mathbf{k})\hat{c}^{\dagger}_{\mathbf{k}\uparrow}\hat{c}_{\mathbf{k}\downarrow} + \alpha^{*}_{\text{SO}}(\mathbf{k})\hat{c}^{\dagger}_{\mathbf{k}\downarrow}\hat{c}_{\mathbf{k}\uparrow}, \tag{3}$$

where $\alpha_{\text{SO}}(\mathbf{k}) = \alpha(\sin(k_y) + i\sin(k_x))$, and $\alpha$ is the Rashba SOC strength. Here, $\hat{c}^{\dagger}_{\mathbf{k}\sigma}$ ($\hat{c}_{\mathbf{k}\sigma}$) creates (annihilates) an electron with momentum $\mathbf{k}$ and spin $\sigma$. We employ the dynamical cluster approximation to treat disorder and SOC within a nonlocal mean-field framework [35, 36]. In DCA, the original lattice is mapped onto a finite-size cluster with $N_c$ sites and periodic

boundary conditions, embedded in a self-consistently determined effective medium. The first Brillouin zone is partitioned into $N_c$ momentum patches, each centered at cluster momentum $\mathbf{K}$, with internal momenta $\tilde{\mathbf{k}}$ such that $\mathbf{k} = \mathbf{K} + \tilde{\mathbf{k}}$ [36]. Due to the presence of Rashba SOC, spin degeneracy is lifted, and the Hamiltonian becomes a $2 \times 2$ matrix in the spin space. The resulting two-band problem has the spin-dependent dispersion,

$$\underline{\varepsilon}_{\mathrm{SO}}(\mathbf{k}) = \begin{bmatrix} \varepsilon_{\mathbf{k}} & \alpha_{\mathrm{SO}}(\mathbf{k}) \\ \alpha_{\mathrm{SO}}^*(\mathbf{k}) & \varepsilon_{\mathbf{k}} \end{bmatrix}, \tag{4}$$

where $\varepsilon_{\mathbf{k}} = -2t(\cos(k_x) + \cos(k_y))$ is the bare tight-binding dispersion.

The DCA self-consistency loop proceeds as follows:

1. **Initialization:** An initial guess for the hybridization function $\underline{\Gamma}_0(\mathbf{K}, \omega)$ is constructed using prior knowledge of the self-energy $\underline{\Sigma}_0(\mathbf{K}, \omega)$ and the cluster Green's function $\underline{G}_0^c(\mathbf{K}, \omega)$:

$$\underline{\Gamma}_0(\mathbf{K}, \omega) = \omega \mathbb{I} - \bar{\varepsilon}_{\mathrm{SO}}(\mathbf{K}) - \underline{\Sigma}_0(\mathbf{K}, \omega) - \underline{G}_0^c(\mathbf{K}, \omega)^{-1}, \tag{5}$$

where $\mathbb{I}$ is the $2 \times 2$ identity matrix and $\bar{\varepsilon}_{\mathrm{SO}}(\mathbf{K}) = \frac{N_c}{N} \sum_{\tilde{\mathbf{k}}} \underline{\varepsilon}_{\mathrm{SO}}(\mathbf{K} + \tilde{\mathbf{k}})$ is the coarse-grained dispersion. If no prior information is available, we set $\underline{\Gamma}_0(\mathbf{K}, \omega) = 0$.

2. **Cluster-excluded Green's function:** The cluster-excluded Green's function is computed as

$$\underline{\mathcal{G}}(\mathbf{K}, \omega) = \left[ \omega \mathbb{I} - \underline{\bar{\varepsilon}}_{\mathrm{SO}}(\mathbf{K}) - \underline{\Gamma}_0(\mathbf{K}, \omega) \right]^{-1}. \tag{6}$$

This is Fourier transformed to real space as

$$\underline{\mathcal{G}}(\mathbf{X}_i, \mathbf{X}_j, \omega) = \frac{1}{N_c} \sum_{\mathbf{K}} \underline{\mathcal{G}}(\mathbf{K}, \omega) e^{i\mathbf{K} \cdot (\mathbf{X}_i - \mathbf{X}_j)}.$$

3. **Disorder sampling:** Disorder configurations $\underline{V}$ are generated stochastically. For each realization, the cluster Green's function is computed via the Dyson equation:

$$\underline{G}^c(\mathbf{X}_i, \mathbf{X}_j, V, \omega) = \left[ \underline{\mathcal{G}}(\mathbf{X}_i, \mathbf{X}_j, \omega)^{-1} - \underline{V} \right]^{-1}. \tag{7}$$

Averaging over many configurations gives the disorder-averaged cluster Green's function $\underline{G}^c(\mathbf{X}_i, \mathbf{X}_j, \omega) = \left\langle \underline{G}^c(\mathbf{X}_i, \mathbf{X}_j, V, \omega) \right\rangle$. This is then Fourier transformed to momentum space to obtain $\underline{G}^c(\mathbf{K}, \omega)$.

4. **Coarse-grained Green's function:** The coarse-grained lattice Green's function is calculated as

$$\bar{\underline{G}}(\mathbf{K}, \omega) = \frac{N_c}{N} \sum_{\tilde{\mathbf{k}}} \frac{1}{\underline{G}^c(\mathbf{K}, \omega)^{-1} + \underline{\Gamma}_0(\mathbf{K}, \omega) - \underline{\varepsilon}_{\mathrm{SO}}(\mathbf{K} + \tilde{\mathbf{k}}) + \underline{\bar{\varepsilon}}_{\mathrm{SO}}(\mathbf{K})}. \tag{8}$$

5. **Hybridization function update:** The hybridization function is updated as:

$$\underline{\Gamma}_n(\mathbf{K}, \omega) = \underline{\Gamma}_o(\mathbf{K}, \omega) + \xi \left[ \underline{G}^c(\mathbf{K}, \omega)^{-1} - \bar{\underline{G}}(\mathbf{K}, \omega)^{-1} \right], \tag{9}$$

where the subscripts "$n$" and "$o$" denote "new" and "old", respectively, and $\xi$ is the mixing parameter. The self-consistency loop is iterated until convergence is achieved, i.e., $\underline{\Gamma}_n = \underline{\Gamma}_o$ within a desired accuracy, which also implies $\underline{G}^c = \bar{\underline{G}}$.

We emphasize that the DCA-SOC method provides a systematic, causal, and self-consistent framework for incorporating nonlocal correlations in disordered systems with spin-orbit coupling. In the limit of $N_c = 1$, it reduces to the coherent potential approximation for disordered lattices with Rashba spin splitting. As $N_c \to \infty$, it converges to a numerically exact solution [35, 36]. Similarly, in the limit $\alpha \to 0$, the formulation recovers the standard DCA for disordered systems without SOC.

# 3 Results and discussion

To explore the interplay between Rashba spin-orbit coupling and disorder in low-dimensional systems, we study a 2D square lattice with binary on-site disorder using the dynamical cluster approximation. The inclusion of Rashba SOC introduces spin-momentum locking, while disorder induces momentum- and site-dependent scattering, together providing a platform to investigate localization physics in the presence of spin-orbit interactions. Unless otherwise stated, all results are shown for a fixed SOC strength of $\alpha = 0.25$ and are compared to the SOC-free case ($\alpha = 0$). To capture the effects of spatial correlations, we perform DCA calculations for finite-size clusters of up to $N_c = 32$ and contrast the results with those from the single-site limit ($N_c = 1$), which corresponds to the CPA. [37, 38, 49]. This comparison allows us to systematically assess the role of nonlocal correlations in shaping the spectral and transport properties of the disordered system.

## 3.1 Spectral properties: Density of states and self-energy

To characterize the influence of Rashba SOC and nonlocal spatial correlations on the electronic structure, we examine spectral properties including the average density of states and the momentum-resolved self-energy. The ADOS provides a global view of the spectral weight distribution, while the self-energy captures quasiparticle lifetimes and the momentum dependence introduced by disorder. Together, these observables allow us to assess the suppression or enhancement of localization effects across disorder and SOC strengths.

The average density of states is calculated according to Eq. (A.1) in Appendix A, and presented in Figure 1 for both $\alpha = 0$ and $\alpha = 0.25$ across different disorder strengths. At weak disorder ($W = 0.20$), the ADOS curves for all the cluster sizes studied in this work overlap, indicating minimal nonlocal corrections in this regime. As disorder increases, the ADOS near the band center, $\omega = 0$, begins to decrease and eventually splits into two subbands due to the binary disorder distribution [37, 38]. A complete gap forms between the subbands at sufficiently high $W$. In the presence of SOC, similar trends are observed at small $W$, but additional features emerge at the band edges. Importantly, Rashba SOC mitigates the impact of disorder: at larger $W$, the disorder-induced gap develops more slowly, and the central ADOS remains higher compared to the $\alpha = 0$ case. This trend is more pronounced for both $N_c = 18$ and $N_c = 32$, and underscores the combined role of SOC and nonlocal spatial correlations in preserving spectral weight near the Fermi level and suppressing localization. At strong disorder ($W = 1.0$), the ADOS exhibits softened tails around the band edges in the presence of SOC, especially for finite-size clusters, due to impurity-induced states captured by the nonlocal correlations. Figure 5 shows the ADOS evolution as a function of SOC strength $\alpha$ for $W = 0.50$ and 1.0. Increasing $\alpha$ reduces the disorder-induced gap, particularly for $N_c = 32$. At stronger disorder ($W = 1.0$), increasing $\alpha$ progressively reduces the ADOS gap, which nearly vanishes for $\alpha = 0.5$ when nonlocal correlations are included. These observations reinforce that the delocalizing influence of Rashba SOC becomes more pronounced in the presence of nonlocal correlations captured by finite-cluster DCA.

To elucidate the influence of Rashba SOC on quasiparticle scattering, we examine the imaginary part of the self-energy computed from Eq. (A.2). Figure 2 displays Im $\Sigma(\mathbf{K}, \omega)$ at representative disorder strengths $W = 0.20$ and $W = 0.80$ for both $\alpha = 0$ and $\alpha = 0.25$, evaluated at three high-symmetry momenta: $\mathbf{K} = (0,0)$, $(\pi, 0)$, and $(\pi, \pi)$. At weak disorder ($W = 0.20$), the self-energy is nearly momentum-independent for both $N_c = 1$ and $N_c = 32$, indicating minimal nonlocal effects. In contrast, at stronger disorder ($W = 0.80$), significant $\mathbf{K}$-dependence emerges in the $N_c = 32$ results, while the $N_c = 1$ case remains featureless, highlighting the importance of spatial correlations captured by DCA [35]. This momentum variation correlates with the disorder-induced spectral broadening observed in the ADOS. Importantly, the

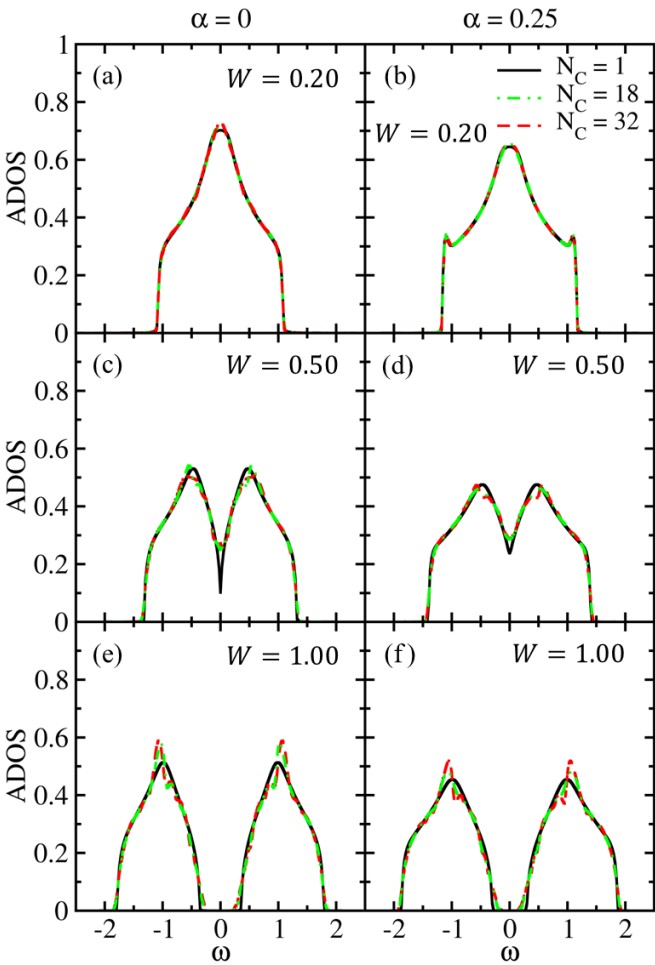

Figure 1: ADOS computed using DCA for disordered two-dimensional electronic systems without Rashba SOC (left panels) and with SOC at $\alpha = 0.25$ (right panels). Panels (a)-(f) compare results for $N_c = 1$ (CPA, black solid curves), $N_c = 18$ (green dot-dashed curves) and $N_c = 32$ (red dashed curves) at increasing disorder strengths: $W = 0.20$ ((a), (b)), $W = 0.50$ ((c), (d)), and $W = 1.00$ ((e), (f)).

inclusion of SOC reduces the **K**-dependent spread of the self-energy for $N_c = 32$, as shown in Figure 2(d), indicating that SOC suppresses spatial inhomogeneity in quasiparticle scattering. From the interference-picture perspective, the momentum-dependent broadening of $\text{Im}\,\Sigma(\mathbf{K})$ without SOC is exactly the signature of anisotropic backscattering rates. Rashba SOC mixes spin channels, averaging out those anisotropies and thereby reducing $\Delta[\text{Im}\,\Sigma]$ across the Brillouin zone, quantifying how spin-orbit interactions homogenize the scattering landscape. A systematic reduction in the momentum-dependence of the self-energy with increasing $\alpha$ is further illustrated in Figure 6.

We further provide a qualitative comparison between our DCA-SOC framework and a commonly used approximation in impurity scattering problems, the self-consistent Born approximation (SCBA) [29, 50–52]. SCBA is valid in the weak-scattering regime near the Fermi level, since it self-consistently sums non-crossing diagrams to give a disorder self-energy [52]. In the context of Rashba SOC in a disordered 2DEG (e.g. Ref. [50]), SCBA predicts a reduction in scattering even without strict time-reversal symmetry, qualitatively in line with our results. However, in the strong disorder regime SCBA fails to incorporate crossing diagrams or spatial nonlocal correlations (vertex corrections) [30, 35], which leads to inaccuracy, especially near band edges. For example, in Ref. [30], an exact numerical evaluation of $\text{Im}\,\Sigma$ reveals soft

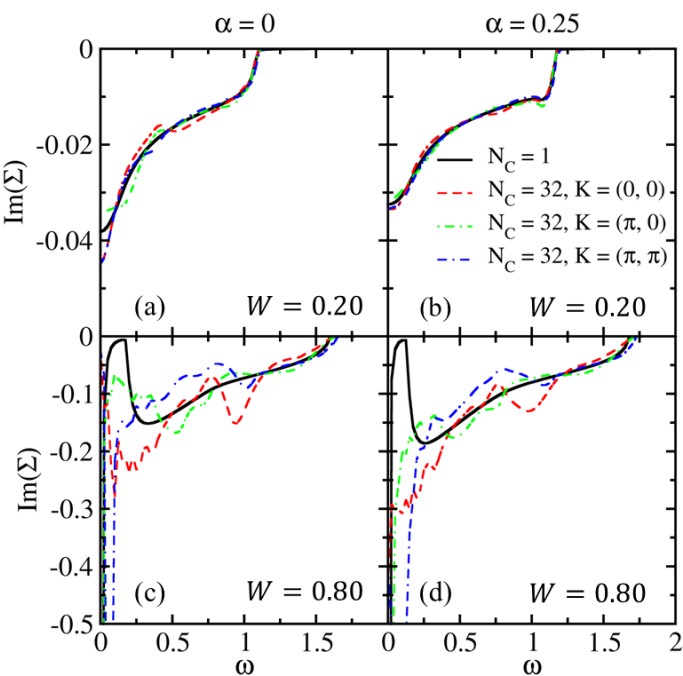

Figure 2: Imaginary part of the self-energy, $\text{Im}\,\underline{\Sigma}(\mathbf{K}, \omega))$, computed using DCA for disordered two-dimensional systems without Rashba SOC (left panels) and with SOC at $\alpha = 0.25$ (right panels). Panels (a) and (b) show results for weak disorder ($W = 0.20$) at $N_c = 1$ (CPA) and $N_c = 32$, evaluated at high-symmetry momenta $\mathbf{K} = (0, 0)$, $(\pi, 0)$, and $(\pi, \pi)$. Panels (c) and (d) show the corresponding results at stronger disorder ($W = 0.80$).

tails that flatten with increasing disorder, but SCBA lacks these features because it neglects nonlocal correlations. In contrast, the coherent potential approximation (CPA) is a single-site mean-field description of disorder and shares the same local (no spatial correlations) limitation as SCBA. As shown in Figure 2, at $W = 0.80$ the imaginary part of the self-energy in the CPA limit ($N_c = 1$) exhibits sharper tails than the results computed for finite-cluster DCA ($N_c = 32$). This demonstrates the ability of our DCA-SOC method to incorporate nonlocal correlation effects that are absent in both CPA and SCBA, especially in the strong disorder limit.

## 3.2 Return probability and delocalization dynamics

Anderson localization is expected to occur in one- and two-dimensional systems for arbitrarily weak disorder [7]. However, spin-orbit coupling (SOC) can suppress localization and promote delocalization by breaking spin-rotational symmetry while preserving time-reversal invariance [1, 8, 17]. To probe this interplay between disorder and SOC, we compute the return probability $P(t)$—the probability that an electron remains at its initial site at time $t$—as defined in Eq. (A.3). Figure 3(a) shows $P(t)$ at $W = 0.50$ for cluster sizes ranging from $N_c = 1$ to $N_c = 32$.

For single-site calculations ($N_c = 1$), $P(t)$ decays rapidly to zero, independent of SOC, indicating delocalized behavior due to the lack of spatial correlations in the single-site system. In contrast, for $N_c > 1$, the inclusion of nonlocal correlations significantly slows the decay of $P(t)$ at $\alpha = 0$, and this effect becomes more pronounced as $N_c$ systematically increases, consistent with the behavior expected in the precursor to localization [35]. When Rashba SOC is introduced, the decay rate of $P(t)$ becomes faster, a direct, time-domain signature of weak an-

tilocalization. In the language of universality classes, the more rapid suppression of the return probability at finite times signals the system's shift into the symplectic class, where destructive interference of time-reversed paths dominates. As shown in Figure 3(b), this behavior persists across a range of disorder strengths. The systematic suppression of $P(t)$ with increasing $\alpha$ reinforces the delocalizing effect of SOC in the presence of disorder. Our DCA-SOC framework not only provides an efficient route to compute single-particle observables but also enables access to the physics of spin-dependent quantum interference in disordered systems. Specifically, by capturing nonlocal correlations and spin-momentum locking, the method allows us to resolve how Rashba SOC promotes delocalization through the suppression of coherent backscattering. This is a hallmark of the symplectic universality class in two dimensions, which is otherwise inaccessible to local approximations like CPA. The reduction in momentum-dependent self-energy and the enhanced spectral weight near the Fermi level reflect the suppression of quantum interference and the emergence of SOC-stabilized diffusive transport channels.

To assess the long-time behavior, we compute the infinite-time return probability $P(t \rightarrow \infty) = p(\eta \rightarrow 0)$, defined in Eq. (A.4). The extrapolation of $p(\eta)$ to zero across all disorder and SOC strengths confirms that the system remains delocalized within the DCA framework. This behavior is consistent with the known limitations of mean-field approaches that lack an explicit order parameter for localization, including DCA [35,40]. As a result, such methods cannot capture the Anderson transition, even at high disorder. Further evidence is provided by the imaginary part of the hybridization function (Figure 7), which remains finite for all $W$ and increases with SOC, indicating persistent coupling to the effective medium and reinforcing the absence of true localization within this framework.

### 3.3 Benchmarking against kernel polynomial method

To assess the accuracy of the extended DCA framework, we benchmark our results against the kernel polynomial method (KPM), a numerically exact approach based on Chebyshev polynomial expansion [45]. The Anderson model with Rashba SOC is implemented using the KWANT package [53], and the KPM calculations are performed on $300 \times 300$ lattices with $10^3$ disorder realizations. Figure 4 presents a comparison of the ADOS computed using DCA ($N_c = 1$ and 32) and KPM across a range of disorder strengths. For $N_c = 32$, the DCA results closely match those of the KPM in both weak and strong disorder regimes. Minor spectral features in the KPM data near $\omega \approx \pm 1$ arise from Gibbs oscillations due to truncation of the polynomial expansion [45]. From a computational standpoint, the DCA calculation with $N_c = 32$ and $2 \times 10^3$ disorder realizations completes in approximately 4 hours, compared to over 30 hours for the corresponding KPM simulations. These results demonstrate that the DCA-SOC framework offers a computationally efficient and scalable alternative to numerically exact methods while reliably capturing key spectral features. Moreover, its mean-field structure provides a natural platform for incorporating spin-orbit coupling, disorder, and strong electronic correlations on equal footing in future studies.

## 4 Conclusion

We have extended the dynamical cluster approximation (DCA) framework to include Rashba spin-orbit coupling (SOC) in the presence of random disorder—an implementation that, to the best of our knowledge, has not been previously reported. Using this approach, we analyzed the density of states, self-energy, and return probability to elucidate the interplay between disorder, SOC, and nonlocal spatial correlations. Our results show that Rashba SOC mitigates disorder effects by suppressing the momentum dependence of the self-energy and delaying the onset of disorder-induced subband splitting in the averaged density of states (ADOS). These

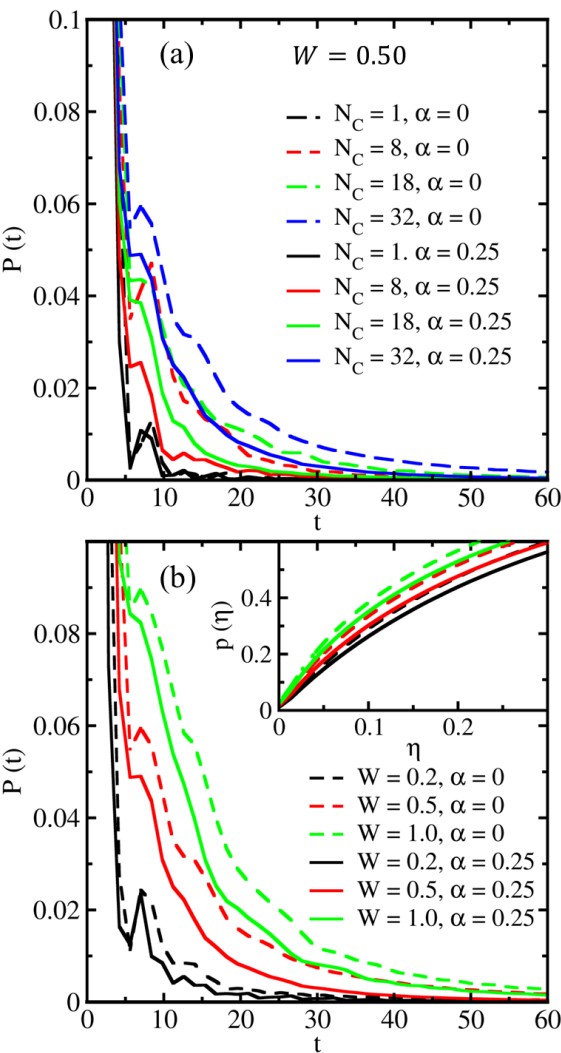

Figure 3: Return probability $P(t)$ as a function of time. (a) Results at fixed disorder strength $W = 0.50$ for cluster sizes $N_c = 1$ (black), $N_c = 8$ (red), $N_c = 18$ (green), and $N_c = 32$ (blue), shown with Rashba SOC ($\alpha = 0.25$, solid curves) and without SOC ($\alpha = 0$, dashed curves). (b) Return probability for fixed cluster size $N_c = 32$ at $W = 0.10$ (black), $W = 0.50$ (red), and $W = 1.00$ (green), again comparing cases with (solid) and without (dashed) SOC. The inset shows the extrapolated infinite-time value $P(t \to \infty) = p(\eta \to 0)$.

trends become increasingly pronounced for larger clusters (up to $N_c = 32$), underscoring the importance of nonlocal correlations. With increasing SOC strength, we observe a systematic enhancement of spectral weight near the Fermi level, signaling SOC-driven delocalization. The return probability exhibits a consistent slowdown in its decay at finite times, reflecting the same delocalizing tendency. At long times, however, the system remains effectively delocalized regardless of disorder strength, reflecting the limitation of DCA as a mean-field theory lacking an order parameter for Anderson localization. While DCA provides a computationally efficient framework for accurate single-particle spectral properties, it cannot capture the critical behavior of the Anderson localization transition (ALT). This limitation can be overcome by typical-medium extensions of DCA, such as the typical-medium dynamical cluster approximation (TMDCA), which has successfully described ALT criticality in both single-band and multiband systems [40, 41, 54]. Our findings demonstrate that the developed DCA-SOC

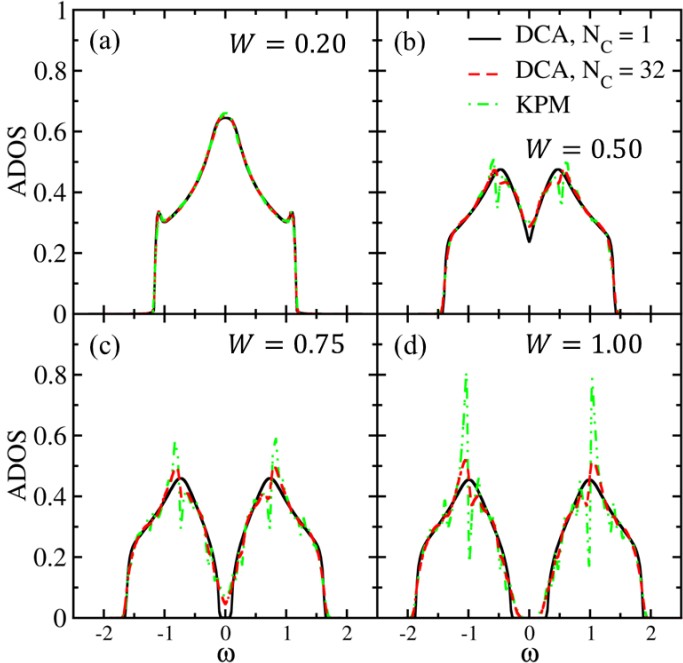

Figure 4: Benchmarking of ADOS computed using DCA for cluster sizes $N_c = 1$ and $N_c = 32$ against results obtained from the KPM for disordered two-dimensional systems with Rashba SOC ($\alpha = 0.25$) at various disorder strengths. KPM calculations were performed on a $300 \times 300$ square lattice with periodic boundary conditions using 1024 Chebyshev moments and $10^3$ disorder realizations for statistical averaging.

method provides direct access to the physics of spin-dependent quantum interference and the symplectic universality class in disordered 2D systems—phenomena that emerge only through the interplay between nonlocal correlations and spin-momentum locking. Building on this foundation, future work will focus on incorporating Rashba SOC into the TMDCA to enable quantitative studies of localization transitions in spin-orbit-coupled, strongly disordered systems. Beyond localization thresholds, the present DCA-SOC framework already enables access to spin-transport phenomena, such as spin Hall and spin relaxation effects, through its self-energy and hybridization functions. Although the infinite-time limit remains delocalized within the mean-field treatment, the finite-time and momentum-resolved observables computed here retain the full imprint of symplectic class physics and weak antilocalization, establishing a foundation for future transport calculations.

## Acknowledgments

Computational resources were provided by the Lehigh University high-performance computing infrastructure.

**Funding information** This research is supported by the U.S. National Science Foundation (NSF) under Award Number DMR-2202101.

# A Single-particle properties and return probability

In our work, to investigate the interplay between disorder and Rashba spin-orbit coupling (SOC) on a 2D square lattice, we compute single-particle properties such as the average density of states (ADOS), the self-energy, as well as the return probability using DCA [35]. In this section, we present the necessary mathematical expressions of the quantities from the self-consistency procedures.

The total average density of states (ADOS) is calculated by taking the trace of the imaginary part of the cluster Green's function, as

$$
\begin{aligned}
\rho_{\text{tot}}^c(\omega) &= \text{Tr}\left\{\underline{\rho}^c(\omega)\right\} \\
&= \text{Tr}\left\{-\frac{1}{N_c}\sum_{\mathbf{K}}\frac{1}{\pi}\text{Im}\,\underline{G}^c(\mathbf{K},\omega)\right\} \\
&= -\frac{1}{\pi N_c}\sum_{\mathbf{K}}[\text{Im}\,G_{\uparrow\uparrow}^c(\mathbf{K},\omega) + \text{Im}\,G_{\downarrow\downarrow}^c(\mathbf{K},\omega)],
\end{aligned}
\tag{A.1}
$$

where, $G_{\uparrow\uparrow}^c(\mathbf{K},\omega)$ and $G_{\downarrow\downarrow}^c(\mathbf{K},\omega)$ represent the cluster Green's function projected to the spin-up and the spin-down electrons respectively.

The disorder self-energy $\underline{\Sigma}$ is extracted using the Dyson equation as

$$
\begin{aligned}
\underline{\Sigma}(\mathbf{K},\omega) &= \underline{\mathcal{G}}(\mathbf{K},\omega)^{-1} - \underline{G}^c(\mathbf{K},\omega)^{-1} \\
&= \omega\mathbb{I} - \underline{\bar{\varepsilon}}_{\text{SO}}(\mathbf{K}) - \underline{\Gamma}_n(\mathbf{K},\omega) - \underline{G}^c(\mathbf{K},\omega)^{-1},
\end{aligned}
\tag{A.2}
$$

where $\underline{\mathcal{G}}(\mathbf{K},\omega)$ is the cluster-excluded Green's function (the Green's function before dressing the disorder).

The return probability, $P(t)$ is a measure of the probability that an electron remains at a given site at time $t$ [55, 56]. By summing over all sites, we calculate $P(t)$ as

$$
P(t) = \left\langle \frac{1}{N_c}\sum_l \left|\frac{1}{2}(G_{\uparrow\uparrow}^c(\mathbf{X}_l,\mathbf{X}_l,t) + G_{\downarrow\downarrow}^c(\mathbf{X}_l,\mathbf{X}_l,t))\right|^2 \right\rangle.
\tag{A.3}
$$

Here, the "1/2" is for normalizing over spin species, so that it preserves $P(t=0) = 1$. $G_{\uparrow\uparrow}^c(\mathbf{X}_l,\mathbf{X}_l,t)$ is Fourier transformed from the frequency-dependent Green's function $G_{\uparrow\uparrow}^c(\mathbf{X}_l,\mathbf{X}_l,\omega)$, so as $G_{\downarrow\downarrow}^c(\mathbf{X}_l,\mathbf{X}_l,t)$ from $G_{\downarrow\downarrow}^c(\mathbf{X}_l,\mathbf{X}_l,\omega)$. It can be shown that in the limit of infinite time, $P(t\to\infty)$, is represented by $p(\eta)$ at $\eta\to 0$:

$$
\begin{aligned}
P(\infty) &= \lim_{\eta\to 0} p(\eta) \\
&= \lim_{\eta\to 0}\frac{-2i\eta}{N_c}\sum_l\int_{-\infty}^{+\infty}\text{d}\omega\text{d}\omega'\left\langle\frac{\bar{A}_l(\omega)\bar{A}_l(\omega')}{\omega-\omega'-2i\eta}\right\rangle,
\end{aligned}
\tag{A.4}
$$

where

$$
\bar{A}_l(\omega) = -\frac{1}{2\pi}[\text{Im}(G_{\uparrow\uparrow}^c(\mathbf{X}_l,\mathbf{X}_l,\omega)) + \text{Im}(G_{\downarrow\downarrow}^c(\mathbf{X}_l,\mathbf{X}_l,\omega))],
\tag{A.5}
$$

is the spectral function calculated from the cluster Green's function for each disorder configuration. Again, "1/2" in Eq. (A.5) normalizes over spin species.

# B Various strengths of Rashba spin-orbit coupling

In the main text, we investigate the role of Rashba SOC in the 2D disordered system for a fixed SOC strength of $\alpha = 0.25$ and various disorder strengths, and compare the results to those

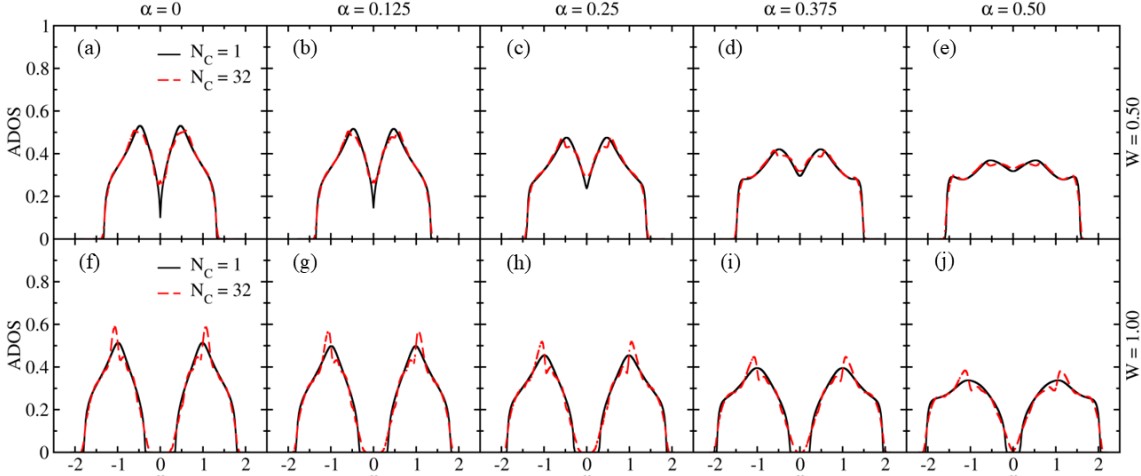

Figure 5: The average density of states (ADOS) calculated using DCA for different values of $\alpha$: $\alpha = 0$ (no SOC, (a), (f)), $\alpha = 0.125$ ((b), (g)), $\alpha = 0.25$ ((c), (h)), $\alpha = 0.375$ ((d), (i)), and $\alpha = 0.50$ ((e), (j)), and two disorder strength: $W = 0.50$ (intermediate disorder strength, in (a)-(e)), and $W = 1.00$ (large disorder strength, in (f)-(j)). We note that panels (a), (c), (f), and (h) restores panels (c), (d), (e), and (f) of Figure 1 in the main text, respectively.

calculated without SOC ($\alpha = 0$). In this Appendix, to understand the interplay of disorder and SOC, we present additional results on the average density of states and self-energy calculated for various SOC strengths: $\alpha = 0$ (absence of SOC), 0.125, 0.25, 0.375, and 0.50 at two disorder strengths, $W = 0.50$ (intermediate disorder), and 1.00 (strong disorder). As in the main text, to understand the implications of nonlocal spatial correlations, we present our results for two cluster sizes, $N_c = 1$ which restores the coherent potential approximation (CPA) results [37, 38, 49], and a finite-size cluster of $N_c = 32$.

The ADOS for different SOC strengths at two representative disorder values, $W = 0.50$, and 1.00 are shown in Figure 5. As discussed in the main text, the role of spatial correlation is important in disordered systems and therefore ADOS for $N_c = 32$ show emerging features compared to those for $N_c = 1$. At $W = 0.50$, as $\alpha$ increases, the spectral weight at $\omega = 0$ systematically begins to increase and eventually the ADOS becomes flattened at $\alpha = 0.50$, indicating that the Rashba SOC mitigates the effect of disorder. At strong disorder ($W = 1.0$) where ADOS has already opened a gap at the band center for $\alpha = 0$, continuously increasing $\alpha$ leads to reduction in the band gap. For $N_c = 32$ where spatial correlations are included, the gap almost vanishes at $\alpha = 0.50$. These observations again support the delocalizing effect driven by SOC in the disordered 2D system.

The imaginary part of the self-energy for different SOC strengths at two disorder values, $W = 0.20$ (weak disorder), and 0.80 (strong disorder) are shown in Figure 6. To understand the momentum dependence of the self-energy, we calculate $\text{Im}\Sigma(\mathbf{K}, \omega))$ at three high-symmetry momenta: $\mathbf{K} = (0, 0)$, $(\pi, 0)$, and $(\pi, \pi)$. For $W = 0.20$, the self-energy shows very small momentum dependence at $\alpha = 0$, and such behavior becomes much weaker as $\alpha$ increases; eventually, all the self-energy curves are almost on top of each other at $\alpha = 0.50$. While the variations of the self-energy with respect to various momenta is greater at $W = 0.80$ in the absence of SOC, and increase of SOC strength from zero to strong value systematically reduces the momentum dependence. Therefore, we again confirm via investigating the self-energy at different SOC strengths that Rashba SOC suppresses the momentum-dependence of the self-energy by reducing the impurity scattering rates.

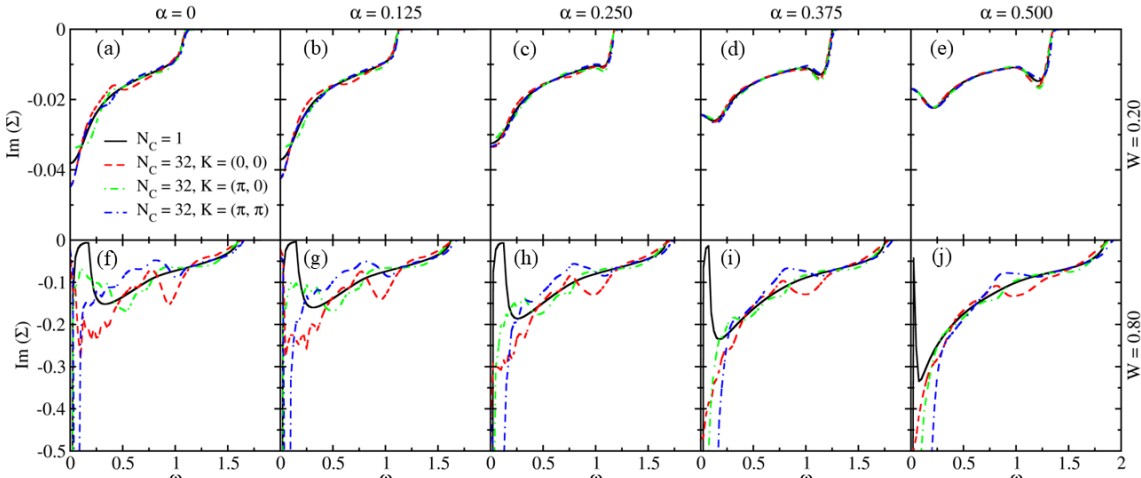

Figure 6: The imaginary part of self-energy $\text{Im}\,\Sigma(\mathbf{K}, \omega)$ for different values of $\alpha$: $\alpha = 0$ (no SOC, (a), (f)), $\alpha = 0.125$ ((b), (g)), $\alpha = 0.25$ ((c), (h)), $\alpha = 0.375$ ((d), (i)), and $\alpha = 0.50$ ((e), (j)). The top panels ((a)-(e)) show the self-energy at $W = 0.20$ (weak disorder), and the bottom panels ((f)-(j)) show the self-energy at $W = 0.80$ (stronger disorder). We note that panels (a), (c), (f), and (h) restores panels (a), (b), (c), and (d) of Figure 2 in the main text, respectively.

## C  Hybridization function

The hybridization function $\underline{\Gamma}(\mathbf{K}, \omega)$ couples the cluster to the host, serving as an effective hopping [35, 36]. The hybridization function is extracted as

$$\underline{\Gamma}(\mathbf{K}, \omega) = \omega \mathbb{I} - \bar{\varepsilon}_{\text{SO}}(\mathbf{K}) - \underline{\Sigma}(\mathbf{K}, \omega) - \underline{G}^c(\mathbf{K}, \omega)^{-1}. \tag{C.1}$$

The delocalization dynamics can be further understood by investigating the hybridization function as well as the return probability which is discussed in the main text. To elucidate the delocalization effect of Rashba SOC in the disordered 2D system, we compute the imaginary part of the hybridization function, integrated over cluster momentum, $\sum_{\mathbf{K}} \text{Im}\,\underline{\Gamma}(\mathbf{K}, \omega)$, at three disorder strengths of $W = 0.20, 0.50$, and $0.80$, and at three SOC strengths of $\alpha = 0$ (absence of SOC), $0.25$, and $0.50$, for $N_c = 1$ and $N_c = 32$. We note that since the hybridization function falls with increasing cluster sizes [36], it becomes much smaller for $N_c = 32$ overall compared to that for $N_c = 1$. In the absence of SOC, a stronger disorder does not suppress the hybridization function; instead, it opens a gap at $\omega = 0$ which becomes wider as $W$ increases. This behavior with respect to increasing $W$ is consistent to that of the ADOS, and the fact that the hybridization function remains finite over the range of frequency where ADOS is finite indicates the absence of localization in 2D, SOC-free disordered system within DCA regardless of the disorder strength. Furthermore, introducing Rashba SOC leads to an increase of the hybridization function for all disorder strength. This leads to an increase of effective hopping between the cluster and the host and can be understood by how Rashba SOC contributes to the bare dispersion relation as in Eq. (4) from the main text. It supports the delocalizing influence of Rashba SOC in the disordered system. We emphasize that the feature of opening a gap at $\omega = 0$ in the hybridization function for $\alpha = 0.25$ and $0.50$ (panels (b) and (c) in Figure 7) is consistent to that observed in ADOS.

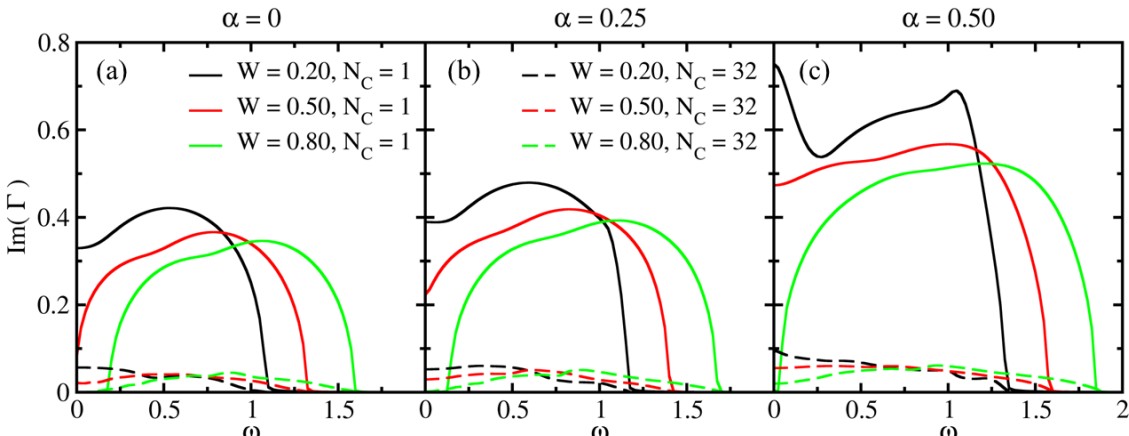

Figure 7: The imaginary part of hybridization function integrated over momentum $\mathrm{Im}(\Gamma(\omega))$ for different disorder strength: $W = 0.20$ (weak disorder), $0.50$ (intermediate disorder), and $0.80$ (strong disorder), and different SOC strength: $\alpha = 0$ (absence of SOC, panel (a)), $0.25$ (panel (b)), and $0.50$ (panel (c)), for $N_c = 1$ and $32$.

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
