# Peer review of "Rashba Spin–Orbit Coupling and Nonlocal Correlations in Disordered 2D Systems"

_SciPost Physics, doi:SciPost Phys. 19, 135 (2025)_

## Round 1 · Referee Report · Anonymous (Referee 1) · 2025-9-27

Report

Referee report on "Rashba Spin–Orbit Coupling and Nonlocal Correlations in Disordered 2D Systems" by Yongtai Li et al.

The interplay between disorder and spin-orbit coupling is a fundamental problem in modern condensed matter physics. Consequently, the development of novel numerical methods to address Anderson localization in the presence of spin-orbit interaction is highly topical. The manuscript under review makes a valuable contribution to this field. The authors successfully extend the dynamical cluster expansion to incorporate the spin-orbit interaction and demonstrate convincing agreement with exact numerical diagonalization.

I find this work interesting and timely. I recommend it for publication in SciPost. However, the manuscript would be significantly strengthened by including comparison with previous analytical results. My specific suggestions are as follows: - A comparison of the calculated average density of states with the self-consistent Born approximation (SCBA) would be highly instructive. One would expect SCBA to provide a reasonable approximation, at least near the band edge, and demonstrating this agreement (or any deviation) would be valuable. - The presentation and discussion of the return probability in Fig. 3 would benefit from a comparison with the simplest analytical models of diffusive propagation. - Similarly, in standard analytical treatments of electron transport, the Cooperon is used to study quantum corrections to the diffusion coefficient and conductivity. Could the authors comment on whether it is possible to extract the Cooperon matrix from their numerical framework? - The authors present results for only two cluster sizes, N_c = 1 and N_c = 32. The convergence of the method at N_c = 32 is not clear. It would be helpful to demonstrate convergence by plotting the ADOS and return probability P(t) for a series of intermediate N_c values.

Recommendation

Ask for minor revision

  • validity: -
  • significance: -
  • originality: -
  • clarity: -
  • formatting: -
  • grammar: -

Author:  Chinedu Ekuma  on 2025-10-13  [id 5919]

(in reply to Report 1 on 2025-09-27)

We thank the referee for a careful reading of our manuscript and for the constructive suggestions. Below, we carefully address each of the points raised.

On comparison with SBCA:
We thank the referee for this valuable suggestion. We have accordingly included a comparison of SCBA results, mainly self-energy, and results from our DCA-SOC formalism. While studies using SCBA have discovered the role of Rashba SOC to be reducing the effect of disorder scattering, which our results are in agreement to, the incapability of incorporating spatial correlations leads SCBA to fail to predict a softened tail in the imaginary part of self-energy that exists in numerically exact studies. Since the coherent potential approximation (CPA), which is a single-site mean-field description of disorder, provides a close correspondence to the SCBA, it does not capture a soft tail in the self-energy that is manifested in finite-size clusters (Nc = 32). Hence, our DCA formalism is capable of capturing nonlocal correlations that were absent in both CPA and SCBA.

On analytic return probability:
We appreciate the referee for this suggestion. We agree that analytical or numerical comparison for the return probability P(t) with diffusive models would be instructive. However, to our knowledge, no reliable numerical results exist for this quantity in disordered systems with SOC. Developing such a comparison requires additional methodological advances, which we plan to pursue in future work.

About the Cooperon matrix:
This is an insightful observation. Our current mean-field formulation does not explicitly compute the Cooperon matrix. Nonetheless, nonlocal correlations beyond the single-site limit are already embedded in the cluster Green’s functions obtained for Nc > 1, effectively capturing the physics that would otherwise emerge through Cooperon-type corrections.

On N_c convergence:
We thank the referee for this valuable suggestion. In response, we have updated the plots of the average density of states (ADOS) and the return probability P(t) to illustrate convergence with cluster size. For the ADOS, since the results for different cluster sizes largely overlap across most frequencies, we now include data for one intermediate cluster size, Nc = 18, which clearly demonstrates this near-convergence behavior. For the return probability, where the curves for different Nc​ values show distinct magnitudes but consistent systematic trends, we have added results for two intermediate cluster sizes, Nc = 8 and Nc = 18, to confirm the robustness and convergence of our findings.

---

## Round 1 · Referee Report · Anonymous (Referee 2) · 2025-10-3

Strengths

  1. Opens a new pathway into a new research direction by studying the combined effects of disorder and Rashba spin-orbit coupling, two aspect that are particularly important, for example, in the context of topological phases.

  2. The paper is written in a clear and intelligible way, and the results are scientifically sound and clearly presented.

Weaknesses

  1. Could make better use of previous literature where the dynamical cluster approximation was extended to systems with Rashba spin-orbit coupling.

Report

This paper addresses the effects of Rashba spin-orbit coupling (SOC) and non-local correlations in disordered 2D systems using a dynamic cluster approximation. While this work does not directly address it, both effects are important in the context of topological phases which can be induced by the Rashba SOC and at the same time destroyed by strong disorder. A fundamental study of the combined effects is therefore interesting and contemporary, and this work open a new pathway in a new research direction, with clear potential for follow-up work.

The paper presents numerical results for the interplay between disorder, SOC, and non-local correlations in a 2D tight-binding model. The main result of this work is that the Rashba SOC counteracts the effects of disorder and drives delocalization. This is interesting, novel, and scientifically sound. I therefore believe that this paper can be recommended for publication.

The only issue to point out is that the statement “We have extended the dynamical cluster approximation to incorporate Rashba spin–orbit coupling” needs better qualification. While this this hasn’t been done before, to the best of my knowledge, in the context of disordered systems, it has been done before in the context of correlated systems and superconductivity in Nagai et al., Phys. Rev. B 93, 220505(R) (2016) (single-site DMFT), Lu and Senechal, Phys. Rev. B 98, 245118 (2018) (clusters), and Doak et al., Phys. Rev. B 107, 224501 (2023) (clusters). Since the basic formalism is the same, the authors should make an effort to acknowledge this.

Requested changes

  1. Add references to previous work which extended the dynamical cluster approximation to systems with Rashba spin-orbit coupling.

Recommendation

Ask for minor revision

  • validity: top
  • significance: high
  • originality: high
  • clarity: top
  • formatting: excellent
  • grammar: excellent

Author:  Chinedu Ekuma  on 2025-10-13  [id 5918]

(in reply to Report 2 on 2025-10-03)

We sincerely thank the referee for the careful and thoughtful review of our manuscript, and for recognizing both the novelty and the broader significance of our work. We are also grateful for the insightful suggestion to acknowledge prior studies that have incorporated Rashba spin–orbit coupling (SOC) in the absence of random defects within dynamical mean-field frameworks.

As correctly pointed out, earlier works such as Nagai et al., Phys. Rev. B 93, 220505(R) (2016); Lu and Sénéchal, Phys. Rev. B 98, 245118 (2018); and Doak et al., Phys. Rev. B 107, 224501 (2023) have indeed extended single-site DMFT/CPA formalism and its cluster extensions (CDMFT) to include Rashba SOC in the context of correlated and superconducting systems. We appreciate this valuable clarification and have revised our manuscript accordingly to properly acknowledge these studies. In particular, we will modify the sentence
“We have extended the dynamical cluster approximation to incorporate Rashba spin–orbit coupling,”
to more accurately reflect that,
“We have extended the dynamical cluster approximation (DCA) framework to include Rashba spin–orbit coupling (SOC) in the presence of random disorder—an implementation that, to the best of our knowledge, has not been previously reported”.

We note that while the cited works focused primarily on superconducting or topological phases in clean or correlated systems, our study targets the interplay between disorder, Rashba SOC, and non-local correlations within the TMDCA formalism, specifically addressing the interplay of spatial correlations, strong electron correlation, Rashba SOC, and random disorder in 2D systems. This distinction underscores the novelty of our approach.

Finally, we agree with the referee that the combination of SOC, electronic correlations, and disorder is highly relevant to understanding possible topological phases. This is something we aim to tackle in the future. Although our current focus is on localization physics, we will explicitly mention in the revised manuscript that incorporating a Zeeman field within our extended TMDCA framework would naturally enable future investigations of disorder- and correlation-driven topological phase transitions in 2D systems.

---

## Round 2 · Referee Report · Anonymous (Referee 2) · 2025-10-16

Report

I believe that the revisions made by the authors sufficiently address the first round criticism, and the revised manuscript can be recommended for publication.

Recommendation

Publish (easily meets expectations and criteria for this Journal; among top 50%)

---

## Round 2 · Referee Report · Anonymous (Referee 1) · 2025-10-17

Report

I am happy with the revisions made by the authors and by their response to my questions. While not all my suggestions were directly incorporated in the manuscript (in particular, for the systems with weak disorder and Rashba coupling the SCBA density of states and return probabilities can be easily found in the literature), I think that current manuscript fully matches the SciPost Physics acceptance criteria.

Recommendation

Publish (meets expectations and criteria for this Journal)

  • validity: -
  • significance: -
  • originality: -
  • clarity: -
  • formatting: -
  • grammar: -

Author:  Chinedu Ekuma  on 2025-10-17  [id 5942]

(in reply to Report 2 on 2025-10-17)

We sincerely thank the referee for their positive assessment and constructive feedback. We are pleased that the revisions have addressed the earlier concerns and appreciate the recommendation for publication.

---

## Round 2 · Author Response

We have carefully addressed each of the comments raised by the reviewers. We appreciated the feedback.

---

## Round 2 · List of Changes

We added some new citations in the manuscript listed below:

Ref 42: Nagai, Yuki et al, Phys. Rev. B 93, 220505 (R) (2016)
Ref 43: Lu, Xiancong et al, Phys. Rev. B 98, 2451118 (2018)
Ref 44: Doak, Peter et al, Phys. Rev. B 107, 224501 (2023)
Ref 51: Langenbuch, M et al, Phys. Rev. B 69, 125303 (2004)
Ref 52: Brosco, V. et al, Phys. Rev. Lett. 116, 166602 (2016)
Ref 53: H. Bruus and K. Flensberg, Introduction to Many-body Quantum Theory in Condensed Matter Physics (Oxford University Press, 2016)

We included in the Introduction section, “While previous studies have incorporated Rashba spin–orbit coupling (SOC) into strongly correlated systems within single-site mean-field approaches such as the DMFT and CPA [42], as well as their cluster extensions [43, 44] to investigate its role in topological superconductivity, to the best of our knowledge, no work has yet examined the competition between random disorder and Rashba SOC using a nonlocal mean-field framework.”

We added at the end of Sec. III A, “We further provide a qualitative comparison between our DCA-SOC framework and a commonly used approximation in impurity scattering problems, the self-consistent Born approximation (SCBA) [29, 51–53]...This demonstrates the ability of our DCA-SOC method to incorporate nonlocal correlation effects that are absent in both CPA and SCBA, especially in the strong disorder limit.”

We adjusted our description of return probability with respect to cluster sizes in Sec. III B, “In contrast, for Nc > 1, the inclusion of nonlocal correlations significantly slows the decay of P (t) at α = 0, and this effect becomes more pronounced as Nc systematically increases…”

We modified the beginning of Sec. IV, “We have extended the dynamical cluster approximation (DCA) framework to include Rashba spin–orbit coupling (SOC) in the presence of random disorder—an implementation that, to the best of our knowledge, has not been previously reported.”

We replotted Fig. 1, the average density of states (ADOS) to include data calculated for Nc = 18. We also modified its caption, “Panels (a)–(f) compare results for Nc = 1 (CPA, black solid curves), Nc = 18 (green dot-dashed curves) and Nc = 32 (red dashed curves) at increasing disorder strengths: W = 0.20 ((a), (b)), W = 0.50 ((c), (d)), and W = 1.00 ((e), (f)).”

We replotted Fig. 3, the return probability to include data calculated for Nc = 8 and Nc = 18. We also modified its caption, “Return probability P (t) as a function of time at fixed disorder strength W = 0.50 for Nc = 1 (black curves), Nc = 8 (red curves), Nc = 18 (green curves), Nc = 32 (blue curves), with Rashba SOC (α = 0.25, solid) and without SOC (α = 0, dashed).”

Minor modifications in the main text:

In Sec. III, we changed the original text “we performed DCA calculations for a finite cluster size of Nc = 32…” to, “we perform DCA calculations for finite-size clusters up to Nc = 32…”
In Sec. III A, we changed the original text “the ADOS curves for Nc = 1 and Nc = 32 overlap…” to, “the ADOS curves for all the cluster sizes studied in this work overlap…”
In Sec. III A, we changed the original text “This trend, more pronounced for Nc = 32…” to, “This trend is more pronounced for both Nc = 18 and Nc = 32…”
In Sec. III A, we changed the original text “the ADOS exhibits softened tails…especially for Nc = 32…” to, “the ADOS exhibits softened tails…especially for finite-size clusters…”
In Sec III B, we changed the original text “Figure 3(a) shows P(t) at W = 0.50 for both Nc = 1 and Nc = 32,” to, “Figure 3(a) shows P(t) at W = 0.50for cluster sizes ranging from Nc = 1 and Nc = 32.”
In Sec. IV, we changed the original text “These effects become more pronounced for larger clusters (Nc = 32),” to “These effects become more pronounced for larger clusters (up to Nc = 32).”

---

## Editorial Decision

published